# NEURALLY AUGMENTED ALISTA

**Freya Behrens**[1,*]**, Jonathan Sauder**[1,*]**, Peter Jung**[1,2,†]
Communications and Information Theory Chair, Technical University of Berlin[1],
Data Science in Earth Observation, Technical University of Munich[2]
{f.behrens,sauder}@campus.tu-berlin.de, peter.jung@tu-berlin.de

## ABSTRACT

It is well-established that many iterative sparse reconstruction algorithms can be unrolled to yield a learnable neural network for improved empirical performance. A prime example is learned ISTA (LISTA) where weights, step sizes and thresholds are learned from training data. Recently, Analytic LISTA (ALISTA) has been introduced, combining the strong empirical performance of a fully learned approach like LISTA, while retaining theoretical guarantees of classical compressed sensing algorithms and significantly reducing the number of parameters to learn. However, these parameters are trained to work in expectation, often leading to suboptimal reconstruction of individual targets. In this work we therefore introduce Neurally Augmented ALISTA, in which an LSTM network is used to compute step sizes and thresholds individually for each target vector during reconstruction. This adaptive approach is theoretically motivated by revisiting the recovery guarantees of ALISTA. We show that our approach further improves empirical performance in sparse reconstruction, in particular outperforming existing algorithms by an increasing margin as the compression ratio becomes more challenging.

## 1 INTRODUCTION AND RELATED WORK

Compressed sensing deals with the problem of recovering a sparse vector from very few compressive linear observations, far less than its ambient dimension. Fundamental works of Candes et al. (Candès et al., 2006) and Donoho (Donoho, 2006) show that this can be achieved in a robust and stable manner with computationally tractable algorithms given that the observation matrix fulfills certain conditions, for an overview see Foucart & Rauhut (2017). Formally, consider the set of $s$-sparse vectors in $\mathbb{R}^N$, i.e. $\Sigma_s^N := \left\{ x \in \mathbb{R}^N \big| \|x\|_0 \leq s \right\}$ where the size of the support of $x$ is denoted by $\|x\|_0 := |\text{supp}(x)| = |\{i : x_i \neq 0\}|$. Furthermore, let $\Phi \in \mathbb{R}^{M \times N}$ be the measurement matrix, with typically $M \ll N$. For a given noiseless observation $y = \Phi x^*$ of an unknown but $s$-sparse $x^* \in \Sigma_s^N$ we therefore wish to solve:

$$\underset{x}{\text{argmin}} \, \|x\|_0 \quad \text{s.t.} \quad y = \Phi x \tag{1}$$

In (Candès et al., 2006) it has been shown, that under certain assumptions on $\Phi$, the solution to the combinatorial problem in (1) can be also obtained by a convex relaxation where one instead minimizes the $\ell_1$–norm of $x$. The Lagrangian formalism yields then an unconstrained optimization problem also known as LASSO (Tibshirani, 1996), which penalizes the $\ell_1$-norm via the hyperparameter $\lambda \in \mathbb{R}$:

$$\hat{x} = \underset{x}{\text{argmin}} \, \frac{1}{2}\|y - \Phi x\|_2^2 + \lambda\|x\|_1 \tag{2}$$

A very popular approach for solving this problem is the iterative shrinkage thresholding algorithm (ISTA) (Daubechies et al., 2003), in which a reconstruction $x^{(k)}$ is obtained after $k$ iterations from initial $x^{(0)} = 0$ via the iteration:

$$x^{(k+1)} = \eta_{\lambda/L}\left( x^{(k)} + \frac{1}{L}\Phi^T(y - \Phi x^{(k)}) \right) \tag{3}$$

---

*equal contribution

†The work is partially funded by DFG grant JU 2795/3 and the German Federal Ministry of Education and Research (BMBF) in the framework of the international future AI lab "AI4EO – Artificial Intelligence for Earth Observation: Reasoning, Uncertainties, Ethics and Beyond" (Grant number: 01DD20001).

where $\eta_\theta$ is the soft thresholding function given by $\eta_\theta(x) = \text{sign}(x) \max(0, |x| - \theta)$ (applied coordinate-wise) and $L$ is the Lipschitz constant (i.e. the largest eigenvalue) of $\Phi^T \Phi$. Famously, the computational graph of ISTA with $K$ iterations can be unrolled to yield Learned ISTA (LISTA) (Gregor & LeCun, 2010), a $K$-layer neural network in which all parameters involved can be trained (each layer $k$ has an individual threshold parameter and individual or shared matrix weights) using backpropagation and gradient descent. LISTA achieves impressive empirical reconstruction performance for many sparse datasets but loses the theoretical guarantees of ISTA. Bridging the gap between LISTA's strong reconstruction quality and the theoretical guarantees for ISTA, ALISTA (Liu et al., 2019) was introduced. ALISTA, introduces a matrix $W^T$, related to the measurement matrix $\Phi^T$ in (3), which is computed by optimizing the generalized coherence:

$$\mu(W, \Phi) = \inf_{W \in \mathbb{R}^{M \times N}} \max_{i \neq j} W_{:,i}^T \Phi_{:,j} \text{ s.t. } \forall i \in \{1, \dots, N\} : W_{:,i}^T \Phi_{:,i} = 1 \tag{4}$$

Then, contrary to LISTA, all matrices are excluded from learning in order to retain desirable properties such as low coherence. For each layer of ALISTA, only a scalar step size parameter $\gamma^{(k)}$ and a scalar threshold $\theta^{(k)}$ is learned from the data, yielding the iteration:

$$x^{(k+1)} = \eta_{\theta^{(k)}} \left( x^{(k)} - \gamma^{(k)} W^T (\Phi x^{(k)} - y) \right) \tag{5}$$

As in LISTA, the parameters for ALISTA are learned end-to-end using backpropagation and stochastic gradient descent by empirically minimizing the reconstruction error:

$$\min_{\theta^{(1)}, \dots, \theta^{(K)}, \gamma^{(1)}, \dots, \gamma^{(K)}} \mathbb{E}_{x^*} \left[ \|x^{(K)} - x^*\|_2^2 \right] \tag{6}$$

The authors rigorously upper-bound the reconstruction error of ALISTA in the noiseless case and demonstrate strong empirical reconstruction quality even in the noisy case. The empirical performance similar to LISTA, the retained theoretical guarantees, and the reduction of number of parameters to train from either $O(KM^2 + NM)$ in vanilla LISTA or $O(MNK)$ in the variant of LISTA-CPSS (Chen et al., 2018) to just $O(K)$, make ALISTA an appealing algorithm to study and extend.

In (Ablin et al., 2019), instead of directly focusing on the reconstruction problem, where $\lambda$ is not known a priori, analytical conditions for optimal step sizes in ISTA are derived for LASSO, yielding Stepsize-ISTA. Stepsize-ISTA is a variant of LISTA in which the measurement matrices are exempt from training like in ALISTA, outperforming existing approaches to directly solving LASSO. Thresholds that are adaptive to the current target vector have been explored in ALISTA-AT (Kim & Park, 2020). Following the majorization-minimization method, component-wise thresholds are computed from previous iterations. In a particular case this yields $\theta_i^{(k)} = 1/(1 + |x_i^{(k-1)}|/\epsilon)$ for some $\epsilon > 0$, known as iterative reweighted $\ell_1$-minimization. By unrolling this algorithm, the authors demonstrate superior recovery over ALISTA for a specific setting of $M, N$ and $s$. In a related approach (Wu et al., 2020) identify undershooting, meaning that reconstructed components are smaller than target components, as a shortcoming of LISTA and propose Gated-LISTA to address these issues. The authors introduce gain and overshoot gates to LISTA, which can amplify the reconstruction after each iteration before and after thresholding, yielding an architecture resembling GRU cells (Cho et al., 2014). The authors demonstrate better sparse reconstruction than previous LISTA-variants and also show that adding their proposed gates to ALISTA, named AGLISTA, it is possible to improve its performance in the same setting of $M, N$ and $s$ as ALISTA-AT.

In this paper, motivated by essential proof steps of ALISTA's recovery guarantee, we propose an alternative method for adaptively choosing thresholds and step sizes during reconstruction. Our method directly extends ALISTA by using a recurrent neural network to predict thresholds and step sizes depending on an estimate of the $\ell_1$-error between the reconstruction and the unknown target vector after each iteration. We refer to our method as Neurally Augmented ALISTA (NA-ALISTA), as the method falls into the general framework of neural augmentation of unrolled algorithms (Welling, 2020; Monga et al., 2019; Diamond et al., 2017). The rest of the paper is structured as follows: we provide theoretical motivation for NA-ALISTA in Section 2, before describing our method in detail in Section 3. In Section 4, we demonstrate experimentally that NA-ALISTA achieves state-of-the-art performance in all evaluated settings. To summarize, our main contributions are:

1. We introduce Neurally Augmented ALISTA (NA-ALISTA), an algorithm which learns to adaptively compute thresholds and step-sizes for individual target vectors during recovery. The number of parameters added does not scale with the problem size.

2. We provide theoretical motivation inspired by guarantees for sparse reconstruction which show that NA-ALISTA can achieve arrive tighter error bounds depending on the target $x^*$.

3. We find that NA-ALISTA empirically outperforms ALISTA and other state-of-the-art algorithms in a synthetic setting as well as in a real-world application from wireless communications and that the gains increase with decreasing $M/N$.

## 2 THEORETICAL MOTIVATION

The thresholds $\theta^{(k)}$ in (5) play an important role in the analysis of ALISTA. While the authors of (Liu et al., 2019) prove that $\theta^{(k)}$ must be larger than a certain value in order to guarantee no false positives in the support of the reconstruction $x^{(k)}$, the thresholds $\theta^{(k)}$ also appear as an additive term in the reconstruction error upper bound.

Thus, to guarantee good reconstruction $\theta^{(k)}$ should be just slightly larger than the value it must surpass in order to both minimize the error and verify the assumption. In this section, we repeat key insights from ALISTA and motivate the choice of adaptive thresholds - the key improvement in our proposed NA-ALISTA. More specifically, we repeat the conditions under which ALISTA guarantees no false positives and highlight an intermediate step in the error bound from (Liu et al., 2019), which tightens when the thresholds can adapt to specific instances of $x^*$.

**Assumption**    (adapted from Assumption 1 from (Liu et al., 2019)[1])
*Let $x^* \in \Sigma_s^N$ be a fixed $s$–sparse target vector. Let $W$ be such that it attains the infimum of the generalized coherence with $\Phi$ (as in (4)) and denote this generalized coherence as $\tilde{\mu} = \mu(W, \Phi)$. Let $s < (1 + 1/\tilde{\mu})/2$. Let $\gamma^{(1)}, \ldots, \gamma^{(K)}$ be any sequence of scalars taking values in $(0, \frac{2}{2\tilde{\mu}s - \tilde{\mu} + 1})$ and $\theta^{(1)}, \ldots, \theta^{(K)}$ with:*

$$\theta^{(k)} \geq \gamma^{(k)} \tilde{\mu} \|x^{(k)} - x^*\|_1 \tag{7}$$

Because in ALISTA, the thresholds $\gamma^{(1)}, \ldots, \gamma^{(K)}$ and stepsizes $\theta^{(1)}, \ldots, \theta^{(K)}$ are optimized in expectation over the training data, the inequality in (7) holds only in the general case if the thresholds are larger than the worst case $\ell_1$-error committed by the algorithm over all training vectors $x^*$ i.e.:

$$\theta^{(k)} \geq \tilde{\gamma}^{(k)} \tilde{\mu} \sup_{x^*} \|x^{(k)} - x^*\|_1 \tag{8}$$

This is needed in order to fulfill the Assumption. Under these conditions it is guaranteed that no false positives are in the support of the reconstruction:

**No false positives**    (Lemma 1 from (Liu et al., 2019))
*Under the settings of the Assumption, it holds that:*

$$\text{supp}(x^{(k)}) \subseteq \text{supp}(x^*) \tag{9}$$

However, the threshold $\theta^{(k)}$ also reappears in the error upper bound. Here we employ an intermediate step of the error upper bound from (Liu et al., 2019):

**Reconstruction error upper bound**    (Theorem 1 from (Liu et al., 2019))
*Under the settings of the Assumption, it holds that:*

$$\|x^{(k+1)} - x^*\|_2 \leq \|x^{(k+1)} - x^*\|_1 \leq \tilde{\mu}\gamma^{(k)}(s-1)\|x^{(k)} - x^*\|_1 + \theta^{(k)}s + |1 - \gamma^{(k)}|\|x^{(k)} - x^*\|_1 \tag{10}$$

Where the first inequality holds for all real vectors and the second inequality is derived in detail in Appendix A of (Liu et al., 2019). According to (10) it is therefore desirable that $\theta^{(k)}$ is as small as

---

[1]Note that in this work and in Liu et al. (2019) the noiseless case is considered to simplify the theorems and proofs. For similar statements in the noisy case, we refer the reader to Chen et al. (2018).

possible, but such that it still satisfies (7). This means that ALISTA has to learn thresholds $\tilde{\theta}^{(k)}$ at least proportional to the largest possible committed $\ell_1$-error over all possible $x^*$ in order to guarantee good reconstruction, for which it is in turn penalized in the error bound.

However, if an algorithm would have access to $\|x^{(k)} - x^*\|_1$ and were allowed to choose thresholds adaptively based on $x^*$, the more relaxed inequality (7) could be employed directly, without taking the supremum over all possible $x^*$ as in (8). Then, this algorithm could obtain a tighter error bound for some individual targets $x^*$ than ALISTA since $\tilde{\theta}^{(k)} \geq \theta^{(k)}$. Finding such an algorithm is the aim of this paper.

## 3 NEURALLY AUGMENTED ALISTA

In order to tighten the error upper bound in (10), we introduce Neurally Augmented ALISTA (NA-ALISTA), in which we adaptively predict thresholds $\theta^{(k,x^*)}$ depending on the current estimate for the $\ell_1$-error between $x^{(k)}$ and the unknown $x^*$. As can be observed from (7), such $\theta^{(k,x^*)}$ must be proportional to $\|x^{(k)} - x^*\|_1$.

In theory, this true $\ell_1$-error could be recovered exactly. This is because there are no false positives in $x^{(k)}$, making it $s$-sparse and for a $\tilde{\mu} < 1/(2s - 1)$ the column-normalized $W^T\Phi$ is restricted-invertible for any $2s$-sparse input (Foucart & Rauhut, 2017) [Corollary 5.4, p.113]. However, it is infeasible to solve such an inverse problem at every iteration $k$. Furthermore, in practice the sparsity is often much larger than what is admissible via the coherence bound. For example, in the experiments of (Gregor & LeCun, 2010; Liu et al., 2019; Wu et al., 2020; Kim & Park, 2020), a sparsity of 50 is used with $M = 250$, $N = 500$. This sparsity already exceeds a maximum admitted sparsity of 11 derived from the minimum theoretical coherence of 0.0447 by the Welch Bound (Welch, 1974), implying that such an exact recovery is not possible in practice anyways.

NA-ALISTA is thus largely concerned with learning for each iteration $k$ a good approximation of $\|x^{(k)} - x^*\|_1$. For this, consider the $\ell_1$-norms of the residual:

$$r^{(k)} := \|\Phi x^{(k)} - y\|_1 = \|\Phi(x^{(k)} - x^*)\|_1 \tag{11}$$

and the iterative update quantity in (5):

$$u^{(k)} := \|W^T(\Phi x^{(k)} - y)\|_1 = \|(W^T\Phi)(x^{(k)} - x^*)\|_1 \tag{12}$$

Both are known to the algorithm even though $x^*$ is unknown. That $r^{(k)}$ and $u^{(k)}$ are useful quantities for approximating the true $\ell_1$-error stems from the fact that $W^T\Phi$ has low mutual coherence, thus being a restricted isometry for sparse vectors. This is visualized in Figure 1. Other useful quantities to approximate the true $\ell_1$-error are given by $\|x^{(0)} - x^*\|_1, \ldots, \|x^{(k-1)} - x^*\|_1$. This is highlighted by Figure 2 and suggests the use of a recurrent neural network in NA-ALISTA. We therefore propose to use an LSTM (Hochreiter & Schmidhuber, 1997) which has two input neurons, receiving $u^{(k)}$ and

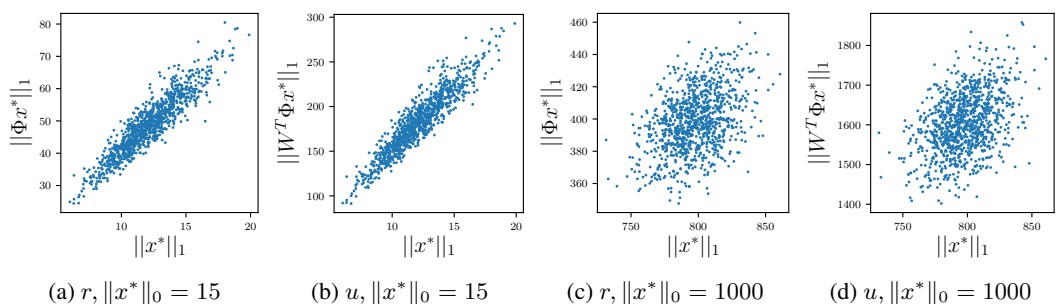

(a) $r, \|x^*\|_0 = 15$  (b) $u, \|x^*\|_0 = 15$  (c) $r, \|x^*\|_0 = 1000$  (d) $u, \|x^*\|_0 = 1000$

Figure 1: Correlation between $\|x^*\|_1$ and $r = \|\Phi x^*\|_1$ or $u = \|W^T\Phi x^*\|_1$. In (a) and (b) for sparse vectors with $\|x^*\|_0 = 15$. In (c) and (d) for non-sparse vectors $\|x^*\|_0 = N$. The non-zero components of $x^*$ are drawn i.i.d. from $\mathcal{N}(0, 1)$ with $N = 1000$. One can see that for sparse $x^*$, $r$ and $u$ are correlated with $\|x^*\|_1$ (Spearman coefficients 0.91,0.92), whereas there is a much weaker correlation for non-sparse vectors (0.38,0.37).

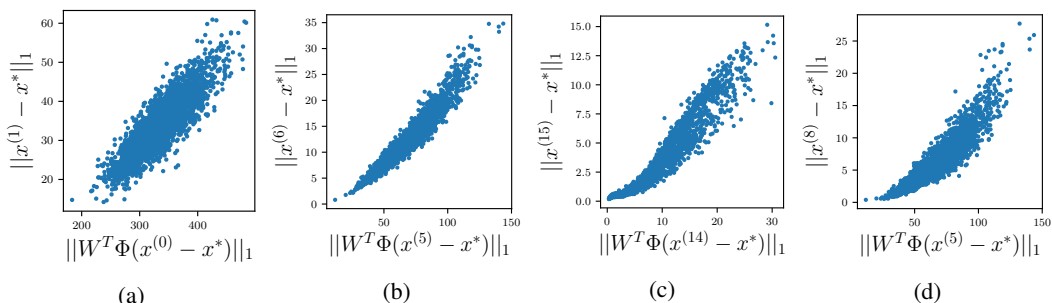

Figure 2: Correlation between $u^{(i)}$ and $\|x^{(j)} - x^*\|_1$ in a trained instance of NA-ALISTA from different iterations $(i, j)$. (a): $(0, 1)$, (b): $(5, 6)$, (c): $(14, 15)$, (d): $(5, 8)$. The Spearman coefficients are $(0.85, 0.96, 0.97, 0.93)$ showing that the strong correlation, is even preserved across multiple iterations (d), suggesting the use of a recurrent neural network to predict $\theta^{(k,x^*)}$. Training was performed with $N = 1000$, $H = 128$ and $K = 16$.

---

**Algorithm 1:** Neurally Augmented ALISTA

---

**Learnable Parameters:** initial cell state $c_0 \in \mathbb{R}^H$, initial hidden state $h_0 \in \mathbb{R}^H$,
LSTM parameters, 1-layer MLP parameters $U_1 \in \mathbb{R}^{H \times H}$, $U_2 \in \mathbb{R}^{2 \times H}$ to map states to outputs.
**Input:** $y$
$x \leftarrow 0$; $h \leftarrow h_0$; $c \leftarrow c_0$
**for** $\{1, \ldots, K\}$ **do**
$\quad r \leftarrow \|\Phi x - y\|_1$
$\quad u \leftarrow \|W^T(\Phi x - y)\|_1$
$\quad c, h \leftarrow \texttt{LSTM}(c, h, [r, u])$
$\quad \theta, \gamma \leftarrow \texttt{Softsign}(U_2(\texttt{ReLU}(U_1 c)))$
$\quad x \leftarrow \eta_\theta\left(x - \gamma W^T(\Phi x - y)\right)$
**end**
**Return** $x$;

---

$r^{(k)}$ at each iteration $k$. This is used to update the internal state and produce the outputs $\theta^{(k,x^*)}$ and $\gamma^{(k,x^*)}$, which are used to compute the next iteration, producing the update rule:

$$x^{(k+1)} = \eta_{\theta^{(k,x^*)}}\left(x^{(k)} - \gamma^{(k,x^*)} W^T(\Phi x^{(k)} - y)\right) \tag{13}$$

A computational expression for NA-ALISTA is given in Algorithm 1. Note that the introduction of LSTM-cells in NA-ALISTA does not significantly increase the required computing power in practice. In fact, in Section 4, we show that small LSTM-cells suffice for best empirical performance, independently of the problem size. Let $H$ be the size of the hidden layer of the LSTM-cells, then the computation for a single forward computation of the cell takes $O(H^2)$ computations. As a regular iteration of ALISTA takes $O(MN)$ operations and computing the $\ell_1$-norm of the update quantity $W^T(\Phi x^{(k)} - y)$ takes an additional $O(N)$ operations, an iteration of NA-ALISTA requires $O(MN + N + H^2)$ operations. For example, when $M = 250$, $N = 2000$, $H = 64$ as in one of the experimental settings in Figure 5, then $H^2/MN = 4096/500000 = 0.008192$, showing that the added computation is negligible in practice.

## 4 EXPERIMENTS

In this section, we evaluate NA-ALISTA in a sparse reconstruction task and compare it against ALISTA (Liu et al., 2019), ALISTA-AT (Kim & Park, 2020), AGLISTA (Wu et al., 2020), as well as the classical ISTA (Daubechies et al., 2003) and FISTA (Beck & Teboulle, 2009). To emphasize a fair and reproducible comparison between the models, the code for all experiments listed is available on GitHub [2].

---

[2] https://github.com/feeds/na-alista

### 4.1 EXPERIMENTAL SETUP

Following the same experimental setup as (Liu et al., 2019; Wu et al., 2020; Chen et al., 2018; Kim & Park, 2020), the support of $x^* \in \mathbb{R}^N$ is determined via i.i.d. Bernoulli random variables with parameter $S/N$, leading to an expected sparsity of $S$. The non-zero components of $x^*$ are then sampled according to $\mathcal{N}(0, 1)$. The entries of $\Phi$ are also sampled from $\mathcal{N}(0, 1)$, before each column is normalized to unit $\ell_2$-norm. $W$ is then computed by minimizing the generalized coherence in (4) between $W$ and $\Phi$ via the Frobenius-Norm approximation using projected gradient descent. This procedure is identical to (Liu et al., 2019; Wu et al., 2020; Kim & Park, 2020). The Adam optimizer (Kingma & Ba, 2015) is used to minimize the $\ell_2$-error from (6) for all algorithms. A test set of 10000 samples is fixed before training and recovery performance is measured with the normalized mean squared error (NMSE):

$$\text{NMSE} = 10 \log_{10} \left( \frac{\mathbb{E}_{x^*}[\|x^{(K)} - x^*\|^2]}{\mathbb{E}_{x^*}[\|x^*\|^2]} \right)$$

A support selection trick was introduced in (Chen et al., 2018) to speed up convergence and stabilize training and has been subsequently used extensively in variants LISTA and ALISTA (Liu et al., 2019; Kim & Park, 2020; Wu et al., 2020). When support selection is used, a hyperparameter $p = (p^{(1)}, \ldots, p^{(K)})$ is set such that for each layer, a certain percentage of the largest absolute values are exempt from thresholding, i.e.:

$$\eta_{(\theta, p^{(k)})}(x)_i = \begin{cases} x_i, & \text{if} \quad |x_i| \geq \lfloor p^{(k)}/N \rfloor \text{-largest value of } |x| \\ \text{sign}(x_i) \max(0, |x_i| - \theta) & \text{else} \end{cases}$$

For a fair comparison, we employ support selection in all learned models compared in this paper similarly to the literature (Liu et al., 2019; Chen et al., 2018; Wu et al., 2020; Kim & Park, 2020). Our AGLISTA implementation follows the description in the paper (Wu et al., 2020): we use exponential gain gates and inverse-proportional-based overshoot gains. The $\lambda$ parameter in ISTA and FISTA was tuned via a grid search, we found that $\lambda = 0.4$ led to the best performance in our tasks. NA-ALISTA by default uses both $r^{(k)}$ and $u^{(k)}$ as inputs to the LSTM in iteration $k$.

When not otherwise indicated we use the following settings for experiments and algorithms: $M = 250, N = 1000, S = 50, K = 16, H = 128$, and $y = \Phi x^* + z$ with additive white Gaussian noise $z$ with a signal to noise ratio $\text{SNR} := \mathbb{E}(\|\Phi x^*\|_2^2)/\mathbb{E}(\|z\|_2^2) = 40\text{dB}$. We train all algorithms for 400 epochs, with each epoch containing 50,000 sparse vectors with a batch size of 512.

### 4.2 COMPARISON WITH COMPETITORS

As an established experimental setting to compare the performance of of ISTA-based methods the compressed sensing, previous work (Liu et al., 2019; Kim & Park, 2020; Wu et al., 2020) has focused on a compression level of $M/N = 0.5$ with sparsity $S$=50 following (Chen et al., 2018). However, practical applications in communication and imaging favor even lower compression rates

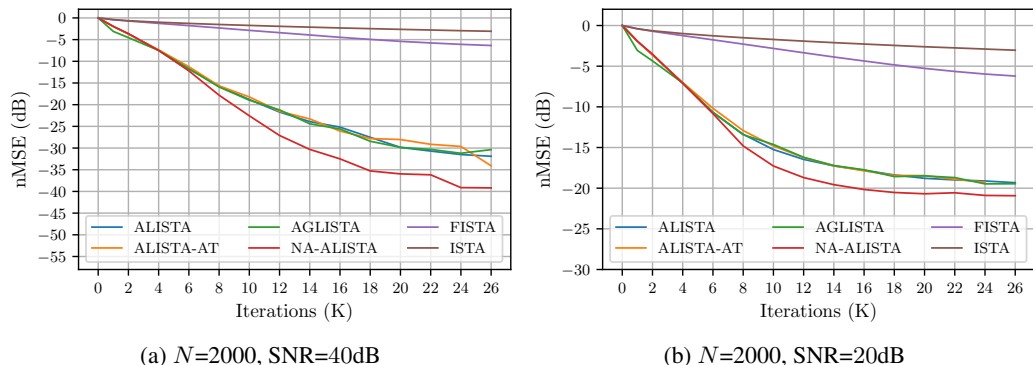

(a) $N$=2000, SNR=40dB                   (b) $N$=2000, SNR=20dB

Figure 3: The reconstruction error for ALISTA, AGLISTA, ALISTA-AT and NA-ALISTA over the number of iterations $K$ for SNR=40dB (3a) and SNR=20dB (3b). NA-ALISTA outperforms all competitors. Results for settings with smaller $N$ can be found in Appendix A.

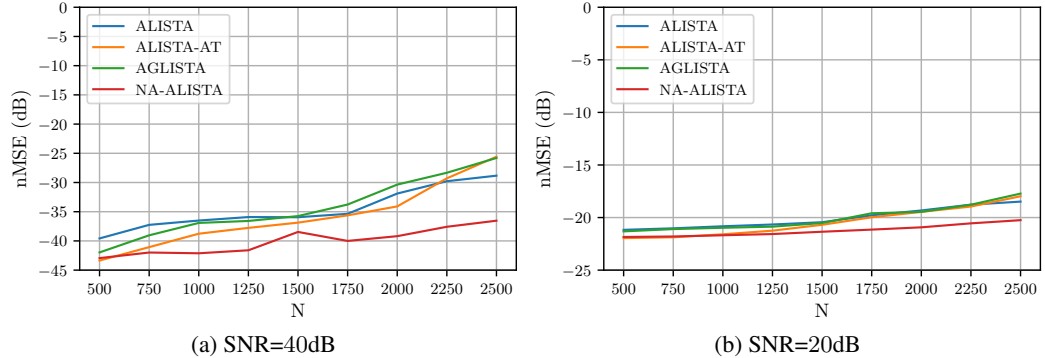

(a) SNR=40dB
(b) SNR=20dB

Figure 4: Reconstruction error over different compression ratios. For a constant expected sparsity of $S = 50$ and $M = 250$ measurements and $K = 26$ iterations, the input size $N$ varies. Both under a SNR of 40dB and 20dB NA-ALISTA increases its reconstruction margin to competitors as $N$ increases and the compression ratio becomes more challenging.

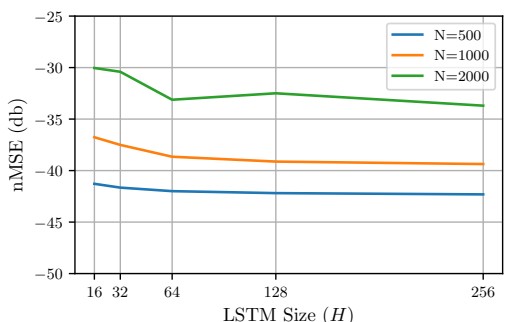 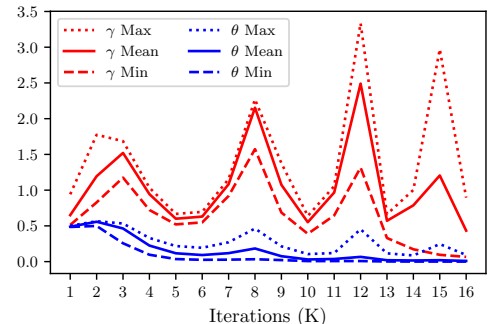

Figure 5: Reconstruction error for varying settings of LSTM size in NA-ALISTA. Larger $N$ profit more from larger $H$, but in all settings an exponential increase of the LSTM size only yields a marginal improvement once $H = 64$ is surpassed.

Figure 6: Predicted step sizes and thresholds from a trained instance of NA-ALISTA ($N = 1000, M = 250, S = 50, K = 16$), highlighting the adpativity of NA-ALISTA. Inference to obtain these values is performed on the test set.

like $10 \ldots 20\%$, which is why we extend our analysis to more challenging rates. To achieve different compression rates we keep the sparsity $S$ and measurements $M$ constant while increasing $N$.

As shown in Figure 3, we first fix $N = 2000$ and observe the reconstruction error for a varying amount of iterations. In Figure 4 we then decrease the compression ratio while keeping the sparsity constant. We observe that NA-ALISTA outperforms state-of-the-art adaptive methods in all evaluated scenarios. Whereas for the more established setting from the literature of $N$=500, the improvement of NA-ALISTA is small, this margin increases as the compression ratio becomes more challenging. In Figure 4a the reconstruction error achieved by ALISTA-AT and AGLISTA deteriorates to the performance of ALISTA, while our NA-ALISTA can sustain its advantage over ALISTA even for compression rates up to 0.1 when $N = 2500$. This suggests that our method is interesting to a wider range of practical applications.

| NA-ALISTA with: | LSTM-$r$ | LSTM-$r, u$ | LSTM-$u$ | MLP-$r, u$ | Vanilla RNN-$r, u$ |
|---|---|---|---|---|---|
| $N = 500$ | -42.00 | **-42.18** | -42.03 | -39.64 | *42.11** |
| $N = 1000$ | -39.15 | -39.12 | -39.24 | -35.64 | **-39.43** |
| $N = 2000$ | **-32.50** | -32.49 | -29.36 | -28.47 | *-24.18** |

Table 1: Reconstruction NMSE in dB for NA-ALISTA with varying inputs ($r^{(k)}, u^{(k)}$) for different neural network architectures with $K = 16$, SNR= 40. For the LSTM it does not seem to matter which quantities we use to estimate the $\ell_1$-error, since all perform equally well. The MLP is outperformed by all recurrent architectures. Even though the Vanilla-RNN can perform on par with the LSTM, it suffers from serious training instability with training resulting in NaNs for $N = 500$ and $N = 2000$ (*) even when trained with a significantly smaller learning rate, which lead us to use the stable LSTM in our experiments.

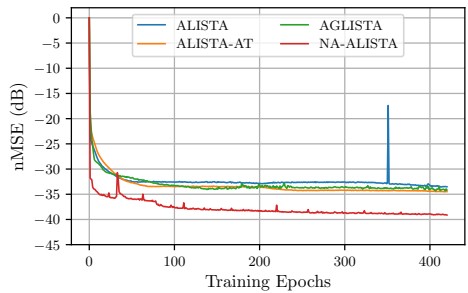
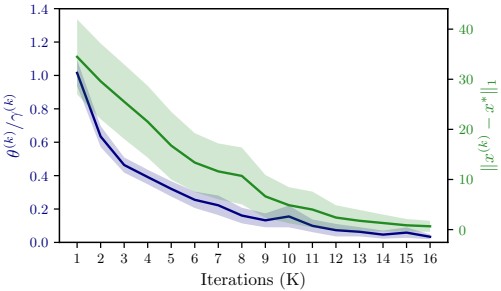

Figure 7: Training curves for $N = 1000$, $K = 16$, SNR$= 40$ from the learned algorithms we compare in this paper, showing that NA-ALISTA outperforms the competitors after only a few epochs of training. Each epoch consists of 50,000 random sparse vectors.

Figure 8: Comparison of the ratio $\theta^{(k)}/\gamma^{(k)}$ with the true $\ell_1$-error $||x^* - x^{(k)}||_1$ at each iteration for NA-ALISTA for the mean of a batch of randomly drawn test data $\{x^*\}$ and its standard deviation. Together these terms behave as desired, see Eq. (7).

To verify that the added computation, determined by the size $H$ of the LSTM, is negligible in practice, we test different settings of $H$. In Figure 5 we show that an exponential increase in hidden neurons yields only a small error reduction for different $N$, suggesting that the size $H = 128$ is a sufficient default value for several settings of $N$. This implies that neural augmentation only marginally affects the runtime. We tested NA-ALISTA using different inputs $r^{(k)}, u^{(k)}$ and architectures in Table 1, justifying the use of an LSTM, and concluding that the all approximations perform similarly and a single approximation of the $\ell_1$-error is sufficient. However, we observe a slight increase in convergence speed and training stability when using both inputs. For a direct comparison of wall clock times we refer the reader to the appendix.

We also evaluate whether the increased empirical performance of NA-ALISTA is truly due to its adaptivity or simply due to its architecture, since the LSTM architecture could in principle enable a more stable optimization of the desired parameters due to more stable gradients. This would imply that when run on a test set, the learned step sizes would not vary depending on the input. Figure 6 shows that this is not the case, since step sizes and thresholds vary within a margin on a test set of 10,000 randomly sampled inputs. Also, the decreasing threshold $\theta^{(k)}$ corresponds to "warm start" behavior for ISTA to first go through a thresholding phase and then through a fitting phase where the threshold becomes essentially zero, see exemplary (Loris, 2009). An additional strength of NA-ALISTA is that it is fast and stable to train, outperforming competitors after only a few epochs, as shown in Figure 7.

As an empirical verification of Assumption 1 in (7) we need to check for every $x^*$, whether the ratio $\theta^{(k,x^*)}/\gamma^{(k,x^*)}$ is proportional to the $\ell_1$-error $||x^* - x^{(k)}||_1$. Since it is infeasible to check the assumption for the infinite set of sparse vectors $\Sigma_s^N$, we empirically verify (7) for a sample of inputs from the training distribution. In Figure 8 the means of both values are proportional to each other for such a test sample, suggesting that the reconstruction bound (10) holds for NA-ALISTA.

### 4.3 REAL DATA SETTING - MULTIPATH CHANNEL ESTIMATION

In this section we evaluate NA-ALISTA for pilot-based multipath channel estimation for *Orthogonal Frequency Division Multiplexing* (OFDM) used in modern wireless networks like LTE and 5G and video broadcasting systems like DVB-T, see e.g. (Tse & Pramod, 2005). Within a scenario-dependent coherence time, the communication channel is almost time-invariant and described by a circular con-

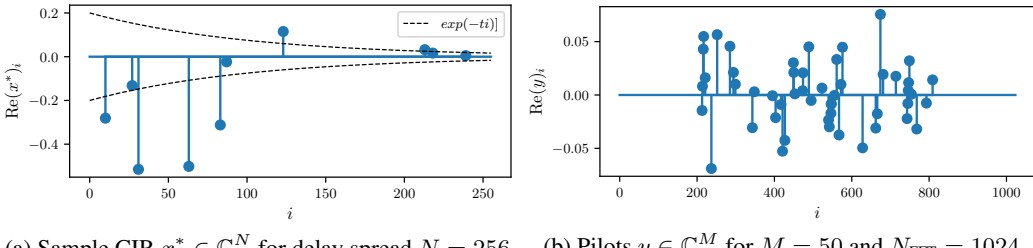

(a) Sample CIR $x^* \in \mathbb{C}^N$ for delay spread $N = 256$    (b) Pilots $y \in \mathbb{C}^M$ for $M = 50$ and $N_{\text{FFT}} = 1024$

Figure 9: Pilot-based multipath channel estimation with compressed sensing. Here, $\Phi \in \mathbb{C}^{M \times N}$ is a subsampled DFT with $N_{\text{FFT}} = 1024$. A random realization of the CIR $x^*$ and the pilot values $y$ are shown.

volution with the channel impulse response (CIR) when using a cyclic prefix. Since this operation is diagonalized by the discrete Fourier transform (DFT), i.e. the information bearing data payload is multiplexed in OFDM onto different frequencies (called subcarriers) and the channel operation in frequency space simplifies to simple multiplications with the DFT of the CIR. Hence, demodulation and decoding of the data message requires accurate knowledge of these instantaneous channel coefficients. In mobile scenarios, the channel is estimated from scattered pilots: on some pilot subcarriers known symbols are transmitted. Conventionally, an equidistant pilot pattern is used as motivated by the Nyquist-criterion, i.e. the maximum delay spread of the channel. Fortunately, the CIR is also often well-approximated by sparse vectors since it concentrates on only few clusters and paths. Due to longer propagation, more delayed components decrease in averaged power. A statistical model like in (Saleh & Valenzuela, 1987) is often used for the power delay profile, i.e. the second order statistics of the random CIR.

From the viewpoint of compressed sensing with partial Fourier matrices (Foucart & Rauhut, 2017), it is known that pilot overhead can be reduced and channel estimation performance improved when penalizing with respect to sparsity and using irregular or random pilot locations (Tauböck & Hlawatsch, 2008; Jung et al., 2009). However, since the iterative compressed sensing algorithms are more complex than linear estimators it is important reach sufficient accuracy in only few iterations and to rely on fast Fourier transforms (FFT). Optimizing generalized coherence for the algorithm via (4) is therefore not feasible since the resulting $W$ can usually not be implemented via FFT.

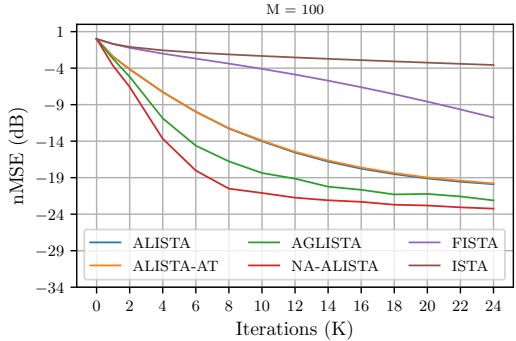

Figure 10: Channel estimation error vs. iterations $K$ for an 10MHz LTE system at SNR=10dB [3]. Amount of pilots $M = 100$, as used in the standard but sampled randomly.

We evaluate NA-ALISTA for a 10MHz LTE system with a FFT size $N_{FFT} = 1024$ with a cyclic prefix of 256 samples ("long CP"), which is capable of dealing with channel delay spreads of up to $N = 256$ (Fig. 9a). A practical system uses $M = 100$ equispaced pilots at distance of 6 in the allowed band of 600 subcarriers for conventional estimation. As motivated above, for compressed sensing we instead use random locations (Fig.9b). It is known that with high probability the partial DFT matrix will have good coherence and thus enable compressed sensing (Foucart & Rauhut, 2017).

As a statistical model for the CIR $x \in \mathbb{C}^N$ we assume an exponentially decaying power delay profile: for every $i \in \text{supp}(x)$ the value $x_i$ is zero-mean complex-Gaussian distributed with variance $p_i = \mathbb{E}|x_i|^2 \simeq \exp(-3.5i)$. The random support itself is uniformly distributed with given sparsity $S = 8$. In Figure 10 we observe that NA-ALISTA finds better solutions with fewer iterations than its competitors and the gain even improves when decreasing number of pilots to 75 (see Appendix).

## 5    CONCLUSION AND FUTURE WORK

In this paper, we propose Neurally Augmented ALISTA (NA-ALISTA), an extension of ALISTA in which the step sizes and thresholds are predicted adaptively to the target vector by a neural network. Besides a theoretical motivation for NA-ALISTA, we experimentally demonstrate that it is able to outperform state-of-the-art algorithms such as ALISTA (Liu et al., 2019), AGLISTA (Wu et al., 2020), and ALISTA-AT (Kim & Park, 2020) in sparse reconstruction in a variety of experimental settings. In particular, NA-ALISTA outperforms the existing algorithms by a wide margin in settings with a large compression.

While in this paper we restrict ourselves to the classical compressed sensing setting, neural augmentation provides a flexible framework for incorporating additional knowledge into classical algorithms. Therefore, an interesting line of future work is to explore how neural augmentation can incorporate notions of structured sparsity or other constraints into sparse reconstruction. There is a plethora of signal processing algorithms, going much beyond variants of compressed sensing and proximal gradient methods, which lend itself to learned unrolling (Monga et al., 2019). Identifying algorithms which could benefit from neural augmentation in the way that ALISTA does is left as future work.

---

[3]For thresholding in the complex case the magnitude of $x_i \in \mathbb{C}$ is changed and the phase remains the same.

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

# A SUPPLEMENTARY EXPERIMENTS ON SYNTHETIC DATA

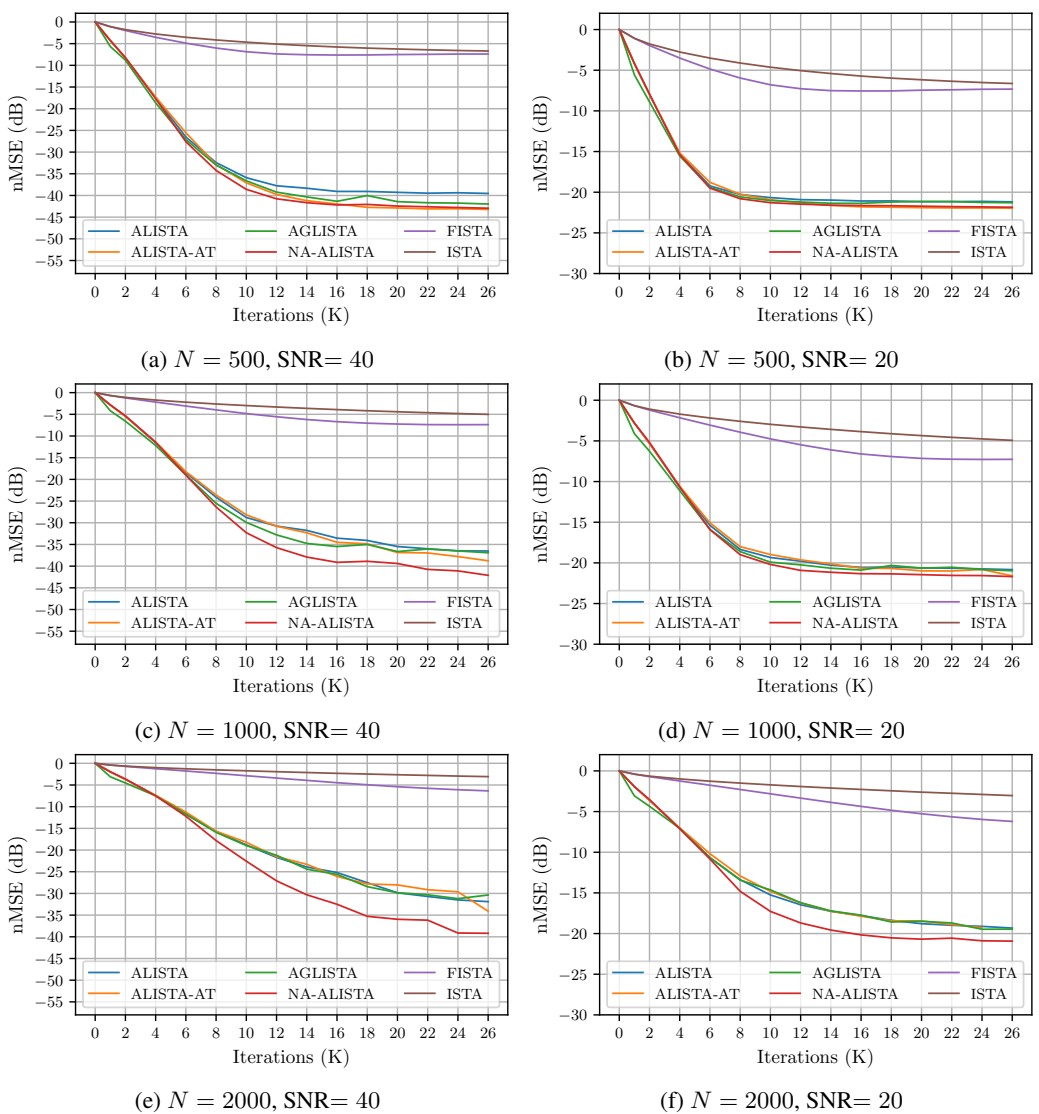

(a) $N = 500$, SNR= 40

(b) $N = 500$, SNR= 20

(c) $N = 1000$, SNR= 40

(d) $N = 1000$, SNR= 20

(e) $N = 2000$, SNR= 40

(f) $N = 2000$, SNR= 20

Figure 11: The reconstruction error for ALISTA, ALISTA-AT and NA-ALISTA over the number of iterations run for different noise and $N$ settings. In 11a, for the standard setting in the literature with $N = 500$ and a noise level of 40dB NA-ALISTA performs on par with competitors after 16 iterations. For an increased $N$=1000 under the same noise level in 11c, our algorithm outperforms the other methods clearly. For a noise level of 20dB all algorithms perform similarly for $N = 500$ and $N = 1000$ and NA-ALISTA outperforms the others at $N = 2000$.

## B COMPARISON OF WALL CLOCK RUNTIMES

In Figure 12, a comparison of the wall-clock time for one iteration NA-ALISTA and ALISTA at inference time for a single batch is shown. A batch size of 5000 instead of 512 is used because otherwise the kernel launch latency dominates the actual computation time for all algorithms when a GPU is used. The results show that NA-ALISTA is feasible to compute and in fact faster than AGLISTA.

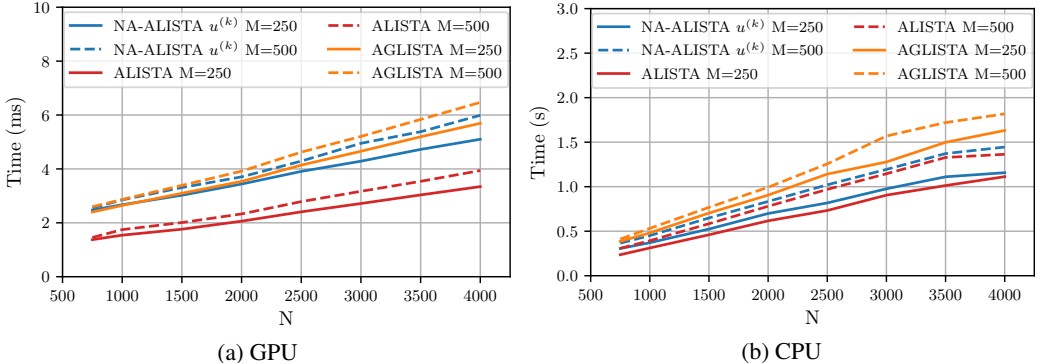

(a) GPU                                    (b) CPU

Figure 12: Wall clock time for a single iteration of NA-ALISTA, AGLISTA and ALISTA with $M$=250 and 500 and $N$ ranging from 750 to 4000 for a single batch of size 5000, averaged over 100 batches. Computations were run on a system with a NVIDIA Tesla P100 GPU and Intel(R) Xeon(R), with the GPU enabled (a) and CPU only (b).

## C SUPPLEMENTARY EXPERIMENTS IN COMMUNICATION SETTING

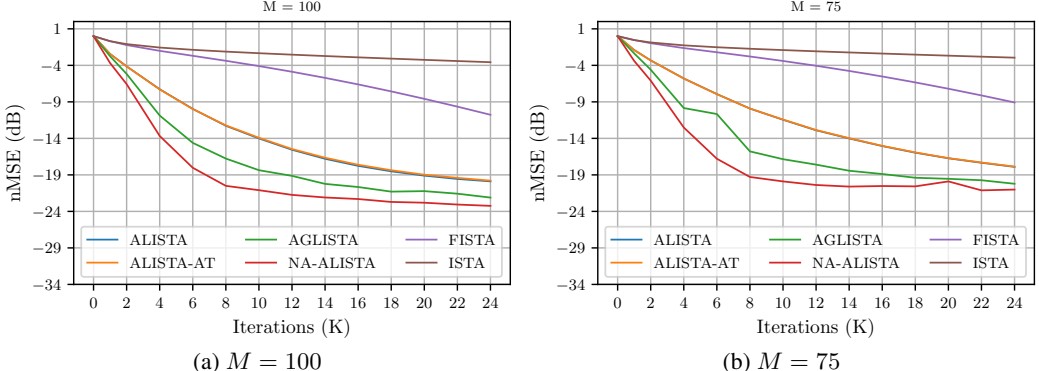

(a) $M = 100$                              (b) $M = 75$

Figure 13: Channel estimation error vs. iterations $K$ for an 10MHz LTE system at SNR=10dB. (a) amount of sampled pilots $M = 100$, while (b) reduces this to $M = 75$, the reconstruction error is still close to using $M = 75$.

