# OpenReview forum: "Neurally Augmented ALISTA"
_ICLR.cc/2021/Conference — ICLR 2021 Poster_

### Official Review · AnonReviewer4 · 2020-10-26
**Official Blind Review #4**

**Rating:** 8
**Confidence:** 4

**Review:**

SUMMARY:
The paper at hand introduces Neurally augmented ALISTA (NA-ALISTA) which is an extension to the previously proposed analytical learned iterative shrinkage threshold algorithm (ALISTA). Both algorithms belong to the class of learned optimization algorithms for solving the compressed sensing problem, i.e., methods that have parameters which are learned via backpropagating through multiple iterations of the algorithm. The key novelty of the NA-LISTA is the LSTM network used to predict thresholds and stepsized used by the algorithm. The experiments show that this adaptive approach improves the performance of ALISTA.

STRENGTHS:
1. After reading the paper I have the impression that the proposed method is thouroughly evaluated. The experimental setup is clear and well-described. Interestingly, the performance of their approach does not depend on wether u^(k) or r^(k) is fed into the LSTM-cell.
2. The results look very promising and the proposed NA-LISTA algorithm seems to consistently outperform the other discussed ISTA variants.
3. The method is well motivated.
4. The paper is very clearly written and well positioned in previously existing literature. All notation is introduced beforehand and it is easy to follow.

WEAKNESSES:
1. The authors claim that the computational time per iteration is not strongly influenced by the forward pass through the LSTM, because of its relatively small architecture. However, I would have liked to see wall-clock time comparison for the different ISTA variants.
2. No completely novel theoretical insights. The lemma and the threorem are adapted from Liu et al. (2019).
3. I find the formulations "An algorithm which approximates such thresholds, resulting in a tighter error bound, is the aim of this paper." somewhat misleading, since there is not tighter bound explicitly stated within the paper.

CONCLUSION:
Overall, I would recommend to accept this submission. The method is solidly justified and the experiments are convincing. I would like to see wall-clock time comparisons in the final manuscript or in the supplemental material, but this is not a major issue in my opinion.

MINOR REMARKS:
- At some points throughout the paper the definition of variables, e.g., N=500, is not written in math mode.
- Also some citations are not correctly formatted, e.g. "Candes, Romberg, Tao and Donoho (Candès et al., 2006; Donoho, 2006)" should be "Candes et al. (2006) and Donoho (2006)".

---

> ### Author Response · Authors · 2020-11-24
> **Re: Official Blind Review #4**
>
> We thank the reviewer for the helpful feedback.
>
> We definitely agree with the reviewer’s wish to include a wall-clock time comparison of the algorithms - this can be found in the appendix of the updated version.
>
> We agree with the reviewer’s assessment that the theoretical insights of NA-ALISTA are largely based on the impressive results from ALISTA. We did our best to highlight their contributions, and to make sure that it is clear that the credit for the Theorem and Lemma belongs to them.
>
> As for the last point, we clarified the somewhat misleading formulation the reviewer pointed out, now saying that we aim to develop an algorithm that can obtain the adaptive version of the error bound which is tighter for some instances of x*, hoping to improve the overall performance.
>
> We have also fixed the citation formatting and the math mode the reviewer pointed out.

---

> > ### Comment · AnonReviewer4 · 2020-11-25
> > **Re: Re: Official Blind Review #4**
> >
> > I thank the authors for their follow-up and I appreciate the inclusion of wall-clock time comparisons in the supplementary material. After reading the updated paper, I chose to update my score to 8.

---

### Official Review · AnonReviewer1 · 2020-10-28
**Improved method for sparse recovery by augmenting ALISTA with good theoretical and empirical motivation**

**Rating:** 8
**Confidence:** 3

**Review:**

### Summary

The paper shows that augmenting analytic learned ISTA (ALISTA) with a small LSTM that predicts step sizes and thresholds improves empirical performance in terms of sparse reconstruction compared to comparable baselines, especially as the compression ratio increases (i.e. ratio of measurement dimension M to sparse vector dimension N).

The proposed method is also nicely motivated by an intermediate step in the ALISTA reconstruction error bounds, where predicting thresholds adaptively given knowledge of the L1 error between estimate at the $k$th step and true target $x^*$ can allow use of a tighter error bound.


### Strong points

S1: Clear explanation of the method in terms of its relation to prior work and theoretical motivation.

S2: Nice empirical exploration of theoretical motivation (i.e. examining correlation between various quantities of interest in Figures 1 and 2)

S3: The approach seems to achieve superior performance compared to comparable baselines.

### Weak points

W1: Only uses synthetic data for evaluation. This is fine to study the properties of the method, and sparse recovery is a general method, but I think the paper would be stronger if the authors also used the approach for some kind of real world task or application. Or at least used parameters for synthetic data that match a real world task, e.g. maybe something from communications. This would also help motivate the method's superior performance for higher compression ratios.

W2: A bit more empirical validation of the theoretical arguments would strengthen the paper. In particular, Figure 8 shows that the ratio of threshold to step size versus iteration $k$ is roughly proportional to the true L1 error, but looking at the bound in equation (7), I wonder if there's a stronger empirical validation, e.g. would it be possible to compute the coherence $\tilde{\mu}$ and check that the ratio is bounded below by coherence times L1 error?

### Recommendation

I think this is a good paper and I recommend acceptance. I would be inclined to increase my score if the weak points I mentioned above were addressed, and depending on answers to my questions below.
The paper is clearly written, the method well-motivated both theoretically and empirically, and achieves superior performance compared to competitive and comparable baselines.

### Questions

Q1: how important is the LSTM architecture? Have you tried other types of RNNs, e.g. vanilla RNN or GRU?

Q2: I was a little confused about the LSTM notation. Does c, h \leftarrow LSTM(c, h, [r, u]) mean that the output is split into these two vectors? A bit more explanation of this notation would be helpful.

Q3: relating to a weak point I mentioned above: is there some motivation for the choice of synthetic data parameters M, N, S, K, H?

Q4: is Figure 3 the mean reconstruction error over the 10k-example test set? Would it make sense to report std devs across examples for each method?

### Other comments:

C1: I found the Figure 1 caption to be a little hard to understand and there's a typo. Adding some punctuation could help: "=15, (a) and (b), and non sparse vectors .., (c) and (d)"

C2: Caption of figure 1 says "whereas there is no obvious correlation for non-sparse vectors.". I wouldn't say there's no obvious correlation just from looking at the scatter plots. For figures 1 and 2, how about measuring and reporting correlation measures, like Pearson and/or Spearman? I suggest Spearman b/c the correlation doesn't seem entirely linear in the Figure 2 plots. Adding these measures would quantify the degree of correlation.

---

> ### Author Response · Authors · 2020-11-24
> **Re: Improved method for sparse recovery by augmenting ALISTA with good theoretical and empirical motivation.**
>
> We thank the reviewer for the detailed review. We address the weaknesses brought to attention by the reviewer as well as their questions in the order posed:
>
> W1: We strongly agree that our paper benefits from a real-world scenario. We have included such an experimental setting in the updated version of the paper.
>
> W2: For the empirical verification of the theoretical results the reviewer suggests to verify the Theorem by measuring the coherence of the matrix and checking if the ratio between theta and gamma is greater than the L1 error multiplied by the coherence as suggested in Equation (6), the difference to our current evaluation being the multiplication by the coherence. We agree that this would give a stronger validation, but note that in our experiment in Fig. 8 the settings are N=2000, K=16, M=250 with a generalized coherence mu=0.31 of Phi and W. Using a sparsity of S=50, the Assumption 1 does not hold anymore since we do not fulfill S < (1 + 1/mu)/2 ~= 2.1. Recovery experiments beyond the coherence bound are also the case in previous literature [2], and interestingly the reconstruction still works despite the large S. In fact, in ALISTA [2], the theorem is also only verified via a correlation instead of strict boundedness.
>
> Q1: We appreciate the concern that there is no ablation study on the effectiveness of using an LSTM in NA-ALISTA. In particular it may not have been clear that a recurrent architecture is necessary at all. In the updated version of the paper, in Table 1, we include a comparison to NA-ALISTA with a simple MLP (i.e. no recurrency) and a Vanilla RNN. We find that the MLP performs worse than all recurrent architectures whereas the Vanilla RNN is able to match the performance of the LSTM. This suggests that recurrent architectures are necessary for good performance. However, the Vanilla RNN exhibits strong training instability - sooner or later it always ends up producing NaNs. In fact we were not able to train a Vanilla RNN for N=2000. Training with a smaller learning rate does not seem to mitigate this. We suspect that the training of a Vanilla RNN would become even more brittle as we increase the number of iterations. This is a known issue, and in fact one of the reasons that the LSTM was introduced [1, Section 1, Paragraph 3 and Section 4.8 ].
>
> Q2: We would like to clarify our notation of the LSTM cell in the pseudo-code block. The LSTM cell in itself is a function of three vectors, a hidden state h_k, a cell state c_k and the inputs, in our case a vector v_k  = [r_k,u_k]. Based on the inputs, it acts on its states to produce a new hidden state h_{k+1} and cell state c_{k+1}, which we denoted as ‘h, c \leftarrow LSTM(h,c,[r,u])’. These new states are then used as the LSTM input in the next iteration. Since generating an output at a specific iteration can differ depending on the application, this step is not part of the general LSTM framework, so we do not capture it in the LSTM function. This closely mimics the Python code for using LSTM cells in PyTorch, where the LSTM cell is a function that return two values (https://pytorch.org/docs/stable/generated/torch.nn.LSTMCell.html)
>
> Q3: The main motivation for the choice of parameters was to allow for a fair comparison between our method and existing literature, which established this choice of parameters. In the real-world scenario we included in the updated version of our paper, these values directly match an application in wireless communication.
>
> Q4: This is correct, we average across a 10k test example set. In ALISTA, the measurements are only taken over a set of 1k examples [2, Section 5.1, first paragraph] which we already increased to get a more stable outcome. The reviewers suggestion to also capture the standard deviation is valid since the reconstruction error indeed varies between different test samples. We evaluated the variance of ||x-\hat{x}|_2/||x||_2 for all methods empirically, but found that it decreases proportionally to the reconstruction error for all of them. Thus we will stick to the common practice of only reporting nMSE without variance in the updated version of the paper.
>
> We addressed the reviewer’s comments by clarifying the formatting of the Fig 1 caption and corroborating our claims by reporting the Spearman correlation measures as suggested.
>
> [1] Hochreiter, Sepp, and Jürgen Schmidhuber. "Long short-term memory." Neural computation 9.8 (1997): 1735-1780.
>
> [2] Liu, Jialin, and Xiaohan Chen. "ALISTA: Analytic weights are as good as learned weights in LISTA." International Conference on Learning Representations (ICLR). 2019.

---

> > ### Comment · AnonReviewer1 · 2020-11-25
> > **Updated review**
> >
> > Thanks to the authors for their responses and their effort in adding additional results and improvements to the paper. I read the revised paper and responses, and I appreciate the ablation versus neural network architecture, as well as the experiment with real-world parameters and its detailed description. I feel that all my comments were addressed, and the paper has been improved. As such I chose to increase my score.

---

### Official Review · AnonReviewer3 · 2020-10-30
**An interesting application of LSTM to ALISTA based on empirical observations**

**Rating:** 7
**Confidence:** 5

**Review:**

This paper extends the framework of ALISTA, a variant of learned ISTA called Neurally Augmented ALISTA (AG-ALISTA), which significantly reduces the number parameters in the model (down to 2 scalars per layer, one for step size and the other for the threshold in soft-thresholding function). Specifically, the authors use a LSTM to generate these two parameters in each layer along iterations, taking reconstruction error related signals as input. This method is based on (1) the previous previous finding of the relation of the step size and threshold with the $\ell_1$ signal recovery error; and (2) the empirical observation of the correlation between the $\ell_1$ signal recovery error and reconstruction error. Experiments in synthetic setting show the superiority of AG-ALISTA over ALISTA and other variants that follow it, especially in settings where the compression ratios are challenging, which is claimed to be more realistic in real-world settings.

Pros:
The most interesting and novel part of this paper is the use of LSTM for the generation of the step size and threshold parameters. The authors use two types of signals related to reconstruction error as the input to the LSTM, which is based on the restricted isometry due to sparsity and observed to be reasonable. And the observation of "not weak" correlation between the $\ell_1$ recovery error and previous reconstruction error also makes it reasonable to use a LSTM structure. These are all interesting observations. Also, the synthetic experimental results do corroborate the effectiveness of the AG-ALISTA model.

Cons:
- Firstly, it is kind of a pity that the authors do not provide some real-world experiments, e.g. compressive sensing, where the compression ratios are challenging indeed. I think this paper would be strong with taht kind of experiments.
- I think eqn (11) and (12) hold when the measurements are noiseless; otherwise they are not exactly accurate. However, noiseless experiments are not presented, nor is there discussion about the noisy/noiseless cases.
- Another concern is that baseline performance of ALISTA with 40dB noises when N=500 seems to be worse than that reported in the previous works. Is it because of the training strategy used? According to the description in this paper, I guess the authors do not use the layerwise training but end-to-end training? This is not clearly stated in the paper.

Others:
- In the last paragraph of page 3, "However, the thresholds that make the error bound tighter vary depending on..." Either "tighter" or "vary" should be deleted.

Overall I think this is an interesting paper. I am willing to further raise my score if the authors address my concerns/questions.

---

> ### Author Response · Authors · 2020-11-24
> **Re: An interesting application of LSTM to ALISTA based on empirical observations.**
>
> We thank the reviewer for their thorough comments.
>
> We have included experiments in a compressed sensing scenario which closely follows a real-world setting.
>
> The reviewer pointed out that the proof from ALISTA [2] we rely on considers the noiseless case. In the LISTA-CPSS paper [1] (by the same authors as ALISTA), specifically Assumption 2 and Theorem 3, properties of desirable sensing matrices to use with LISTA in the presence of noise are proven. This proof of ALISTA [2] very closely follows the proof from LISTA-CPSS, but instead considers the noiseless case “for simplicity of the proofs” (Page 3, last paragraph), instead showing resilience against noise empirically. We added a comment as a footnote in the revised version of our paper in Section 2. The proofs of AGLISTA [3] and ALISTA-AT [4] are in turn are based on these noiseless proofs of ALISTA, but all also provide experiments only for the noisy setting. We suspect that the reason for these two essential papers not providing experiments in the noiseless setting is that for sparse signals with random normal elements, NMSE below -50 dB essentially becomes indistinguishable from numerical errors.
>
> The fact that the baseline performance of ALISTA with 40dB noise and N=500 is slightly worse than reported by the original authors may stem from two things: learning rate schedule and the way noise is applied. In ALISTA, an exponential decay of the learning rate is employed, whereas we train with a smaller but constant learning rate. We believe that employing such a scheme and training for longer could improve the performance of all algorithms by a small margin in the SNR=40 case. As for the noise: in ALISTA,  and subsequent works, the standard deviation of the noise is estimated for each dimension of the vectors per batch: https://github.com/VITA-Group/ALISTA/blob/master/utils/train.py#L84-L88. This means that the fluctuation in SNR for each individual target vector depends on the batch size. In our implementation, noise is applied instantaneously, leading to each vector having exactly the prescribed SNR. We do not believe that the difference stems from a layer-wise training procedure: in fact the authors of ALISTA explicitly state that they use end-to-end training [2, page 5, below eq. 16] for the sparse reconstruction experiments. The layer-wise joint training refers to their convolutional sparse coding experiment. In any case, we are confident our paper allows for a fair comparison between the compared algorithms, as all algorithms are trained in exactly the same setting.
>
> We have also fixed the typo that has been brought to attention. (We clarified the complete paragraph in the new version.)
>
> [1] Chen, Xiaohan, et al. "Theoretical linear convergence of unfolded ISTA and its practical weights and thresholds." Advances in Neural Information Processing Systems (NeurIPS). 2018.
>
> [2] Liu, Jialin, and Xiaohan Chen. "ALISTA: Analytic weights are as good as learned weights in LISTA." International Conference on Learning Representations (ICLR). 2019.
>
> [3] Wu, Kailun, et al. "Sparse Coding with Gated Learned ISTA." International Conference on Learning Representations (ICLR). 2019.

---

> > ### Comment · AnonReviewer3 · 2020-11-24
> > **Re: Re: An interesting application of LSTM to ALISTA based on empirical observations.**
> >
> > Thank you for your response, clarifications and the efforts for the new wireless data experiments. It a great job and will be a good complement to the paper. Therefore, I raised my score.

---

### Official Review · AnonReviewer2 · 2020-10-31
**A reasonable improvement over LISTA with somewhat insufficient experiments.**

**Rating:** 5
**Confidence:** 4

**Review:**

This paper adds LSTM to adjust the step size and threshold for LISTA, an optimization algorithm of sparse regression problems.

Using LSTM to dynamically determine those optimization parameters seems to be reasonable.
Still, I found the experiments to be a bit insufficient to convince me of its improvement over LISTA or even most vanilla solvers: Figure 3,4,5 all showing MSE to some iterations when only one optimizer reaches its optimality, or even not one reaching optimality. I also think the other algorithms, with the step size tuned better, could be reaching faster convergence, however, the paper fails to mention how the parameters are tuned for those methods. On top of this, no mentions of the actual running time of the optimizer as LSTM could be really slow.

small questions:

What is the point of the study showed in figure 1&2 as by definition,  as $W$ is already supposed to be an isometric mapping?

---

> ### Author Response · Authors · 2020-11-24
> **Re: A reasonable improvement over LISTA with somewhat insufficient experiments.**
>
> We thank the reviewer for their feedback.
>
> We agree with the reviewer that the number of iterations was insufficient to convincingly convey that NA-ALISTA is indeed better - we increased the number of iterations in the updated version of the paper. We also include an analysis of the actual running time in the Appendix.
>
> As for tuning the other algorithms: In FISTA and ISTA, only lambda has to be tuned, which we did via a grid search (as mentioned on Page 6, Paragraph 3). ALISTA is also rather straightforward to tune: the support selection hyperparameter p has a large effect, and we confirmed that the value proposed by the authors [1] leads to the best performance. Beyond that we found that as long as the initial thresholds aren’t too big to clip all values, ALISTA always reached the same reconstruction performance after enough training. In ALISTA-AT, there is only one more hyperparameter over ALISTA, an epsilon to prevent division by zero. We use the authors’ recommended value. Tuning AGLISTA is a bit more nuanced as another 3 initial learnable parameters have to be tuned: we tuned these per hand to the best of our ability. Note that these are only initial parameters and that stochastic gradient-based learning should lead them to at least a local optimum.
>
> To address the question about Figures 1 & 2: Indeed Figure 1 (a) and (b) are redundant with the text. Figure 1 (c) and (d) shows that W is not isometric for dense vectors, which directly follows from M<<N. Thus, it is correct that everything in Figure 1 can be inferred from the text - nonetheless we believe the Figure increases readability and thus leave it in the paper. In fact, Reviewer 3 specifically mentioned this.
>
> On the other hand, Figure 2 shows something much less obvious: even though the matrices involved are an isometry for sparse vectors, this does not necessarily mean that vector norms are preserved through an iteration of the algorithm, i.e. a gradient step with a dynamically computed step size followed by the l1 proximal operation. This relationship, and the fact that it is even preserved across multiple iterations are highlighted by Figure 2.

---

### Author Response · Authors · 2020-11-24
**Rebuttal Revision**

We thank reviewers for their comments, all of which provided helpful suggestions to improve the paper. We agree with the comments and did our best to incorporate their feedback. Before addressing each reviewer’s comments individually, we summarize the main improvements of the paper during this author response phase:


### Real world setting
To some of the reviewers, the choices of problem size M, N, S, and iterations K seemed arbitrary, leading them to wish for a real-world setting. In the updated version of our paper, we have included a setting with real problem sizes from pilot-based multipath channel estimation in wireless communication based on Orthogonal Frequency Division Multiplexing. In this setting, it is particularly important for reconstruction algorithms to require few iterations and use fast transforms (i.e. FFT). Specifically, we focus on an LTE setting with N=256, M=100, S=8 using exponentially decaying power delay profiles. Our method, NA-ALISTA, also performs better than all other evaluated algorithms in this setting. As we make the compression ratio more difficult and set the number of pilots to M=75, the gap by which competitors are outperformed widens.
### Running until convergence
Reviewers have noted that in some of our evaluated settings, setting the number of iterations K to 16 is not sufficient to see which algorithm reaches the best reconstruction. We agree: in the updated version we have significantly increased the number of iterations for which we show sparse reconstruction results up to K=26.
### Wall Clock time
Multiple reviewers noted that our claim that we do not significantly increase computational cost is not sufficiently backed up. The reviewers wished to see a wall clock time comparison of the algorithms. We definitely agree and have included such an analysis in the appendix of the updated version of our paper, corroborating our claims.


We kindly ask the reviewers to have a look at the updated paper, hoping that we could address their questions and uncertainties & incorporate their feedback to improve our paper.

---

### Decision · Program_Chairs · 2021-01-07
**Final Decision**

**Decision:**

Accept (Poster)

**Comment:**

The reviewers and AC liked the basic idea of how this paper improves on ALISTA, and the initial scores were high. Because the contributions rely quite a lot on empirical demonstrations, the reviewers asked for more experiments, changes to experiments, and timing results. The revision and rebuttal addressed most of these requests.  The multipath channel estimation problem was interesting though outside the scope of the AC and reviewer's expertise, so it is hard to evaluate how helpful the method is in that particular setting.